# Accelerating 3D genomics data analysis with Microcket

Yu Zhao[1], Mengqi Yang[2,3], Fanglei Gong[2], Yuqi Pan[2,4], Minghui Hu[1], Qin Peng[5], Leina Lu[6], Xiaowen Lyu [7] & Kun Sun [2] ✉

The three-dimensional (3D) organization of genome is fundamental to cell biology. To explore 3D genome, emerging high-throughput approaches have produced billions of sequencing reads, which is challenging and time-consuming to analyze. Here we present Microcket, a package for mapping and extracting interacting pairs from 3D genomics data, including Hi-C, Micro-C, and derivant protocols. Microcket utilizes a unique read-stitch strategy that takes advantage of the long read cycles in modern DNA sequencers; benchmark evaluations reveal that Microcket runs much faster than the current tools along with improved mapping efficiency, and thus shows high potential in accelerating and enhancing the biological investigations into 3D genome. Microcket is freely available at https://github.com/hellosunking/Microcket.

In cell biology, the three-dimensional (3D) genome architecture instructs pivotal regulatory information underlying cellular differentiation, development, and diseases[1–4]. To gain deeper insights into the structure and dynamics of genome conformation, international consortia like the 4D Nucleome Network (4DN)[5] have been launched, aiming to provide the spatial and temporal information of genome architecture and understand how it relates to genome function. Experimentally, various powerful technologies have been widely applied to map 3D genome architecture in diverse contexts, including high-throughput chromosomal conformation capture (Hi-C), and emerging Micro-C approaches[6–8]. Analysis of the data generated from these 3D genome experiments consists of two major sections: sequencing read alignments and pairs inferences (upstream), as well as functional interpretations and data visualizations (downstream). Currently, the downstream analyses (e.g., TAD identification) are relatively efficient with various well-developed and dedicated tools, e.g., Juicer[9]. However, as resolving the finer-scale chromatin organization often requires an extremely large volume of data (e.g., samples in 4DN project are sequenced to billion-level reads), although various tools had been developed[9–12], the upstream analyses are still challenging and time-consuming (may take weeks to months[9]). In addition, emerging protocols[7,13] would generate a higher proportion of reads with more than two interacting fragments, which are not supported by most current tools; hence, development and/or update of computing tools are always needed

to meet the requirements raised from advancements in sequencing technology.

To accelerate the analysis of large-scale 3D genome data, in this study, we developed Microcket with a focus on rapid read alignment and pairs extraction (Supplementary Figs. S1, S2). We benchmarked Microcket against the current tools using both simulated and real-world data, and the results demonstrated advantages in both mapping speed and efficiency of Microcket.

## Results

### Schematic workflow of Microcket

The workflow and main features of Microcket are shown in Fig. 1, Table 1, and Supplementary Fig. S1. Unlike most current tools that map the raw reads directly, Microcket first utilizes a preprocessing procedure to get rid of sequencing adapters, low-quality cycles, as well as PCR duplicates to exempt the alignment cost of these unfavorable reads. Afterwards, to take advantage of the shortened library size in emerging Micro-C protocols while lengthened read cycles in current sequencers, Microcket attempts to stitch the preprocessed reads before actual alignment: for a DNA fragment whose size is shorter than the paired-end read length, the tailing cycles in read 1 and 2 should be complementary to each other therefore allowing one to merge the two paired-end reads into a single stitched fragment. By doing so, paired-end reads where both read 1 and 2 contain the interaction sites (i.e., chimeric

[1]Molecular Cancer Research Center, School of Medicine, Shenzhen Campus of Sun Yat-sen University, Sun Yat-sen University, Shenzhen 518107, China. [2]Institute of Cancer Research, Shenzhen Bay Laboratory, Shenzhen 518132, China. [3]Department of Chemical and Biological Engineering, Division of Life Science, Hong Kong University of Science and Technology, Hong Kong SAR 999077, China. [4]Department of Biology, School of Life Sciences, Southern University of Science and Technology, Shenzhen 518055, China. [5]Institute of Systems and Physical Biology, Shenzhen Bay Laboratory, Shenzhen 518132, China. [6]Department of Genetics and Genome Sciences, School of Medicine, Case Western Reserve University, Cleveland, OH 44106, USA. [7]State Key Laboratory of Cellular Stress Biology, Fujian Provincial Key Laboratory of Reproductive Health Research, Fujian Provincial Key Laboratory of Organ and Tissue Regeneration, School of Medicine, Faculty of Medicine and Life Sciences, Xiamen University, Xiamen 361102, China. ✉e-mail: sunkun@szbl.ac.cn

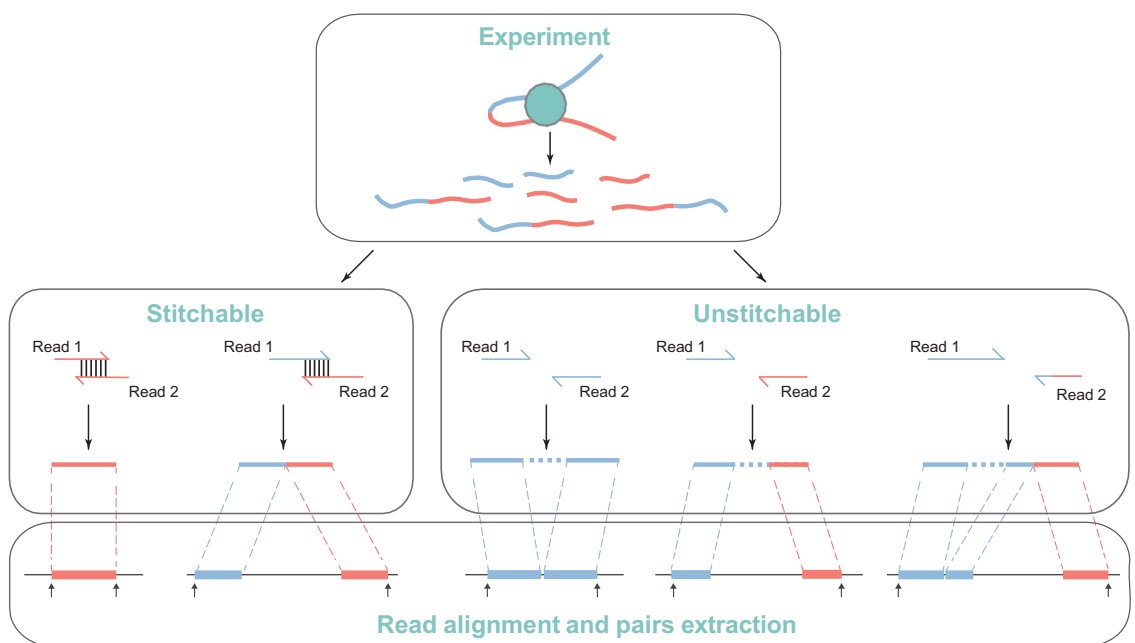

**Fig. 1 | The read stitching and pairs extraction strategies in Microcket.** Paired-end reads that are shorter than twice of the sequencing cycles were identified and stitched into single-end fragments; after alignment of both stitched and unstitched reads, Microcket kept those that could be inferred as one or two segments and reported the outmost ends as pairs.

## Table 1 | Major features of Microcket and the current tools

| | Supported data type | Preprocessing | Aligner | Output format | Reference |
|---|---|---|---|---|---|
| Microcket | Hi-C/Micro-C | Yes | BWA/STAR | Pairs/hic | This study |
| Distiller | Hi-C/Micro-C | No | BWA | Pairs | NA |
| FAN-C | Hi-C only | No | BWA/Bowtie2 | Other | Kruse et al.[10] |
| HiC-Pro | Hi-C/Micro-C | No | Bowtie2 | Other | Servant et al.[12] |
| HiCUP | Hi-C only | Yes | Bowtie2 | Other | Wingett et al.[11] |
| Juicer | Hi-C/Micro-C | No | BWA | Hic | Durand et al.[9] |

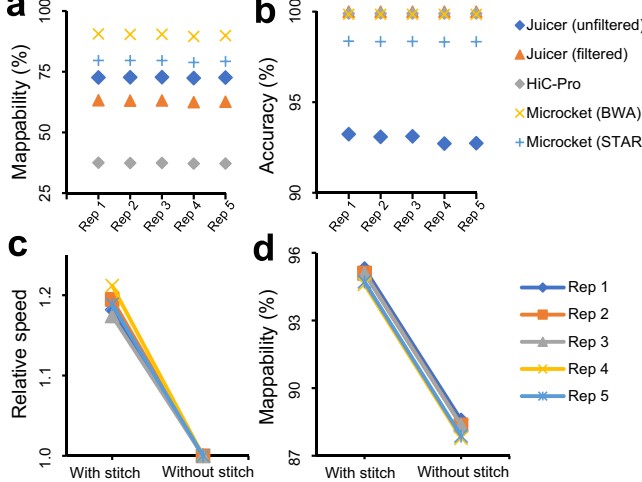

**Fig. 2 | Benchmark evaluations on in silico simulation data. a** Overall mappabilities and **b** accuracies of Juicer, HiC-Pro, and Microcket. For Juicer, it provided raw and filtered results using a mappability score of 30. **c** Relative alignment speed and **d** mappabilities on stitchable reads processed by Microcket (with BWA) with stitching strategy or not. The experiment was repeated 5 times.

reads, which is one of the most challenging scenarios for alignment) are converted into long single-end fragments with interaction sites preserved in the middle. As chimeric reads could not be mapped to the genome normally, the aligners need to recognize the chimeric sites and split the reads into two segments for further alignment, which would spend additional time and reduce the mapping efficiency if the segments after splitting are too short to get uniquely mapped. Hence, read stitching could reduce the tough chimeric reads from two (i.e., read 1 and read 2) to one (i.e., the stitched fragments, which are usually longer than the original reads), and therefore would improve both the mapping speed and sensitivity. In addition, besides the commonly used BWA[14] software, Microcket supports STAR[15] as an optional aligner; Microcket further implements in-house programs (in C++) for rapid alignment filtering (e.g., remove low-quality or incomplete alignments) and pair extraction. Lastly, Microcket records the pairs in mainstream *pairs*, *hic* and *cool* formats, allowing seamless integrations with various downstream tools, e.g., filtering of low-quality pairs, masking regions with mapping issues, as well as functional annotations[9,16–18].

## Benchmark evaluation using in silico simulations

To assess the performance of Microcket, we first performed an in silico simulation to evaluate the accuracy of Microcket with different aligners (i.e., BWA or STAR) versus Juicer[9] and HiC-Pro[12], two of the most widely used software in this task. As shown in Fig. 2 and Supplementary Figs. S2, S3, Microcket generated significantly more pairs than both Juicer and HiC-Pro (Fig. 2a; all $P < 10^{-6}$ for comparing Microcket (with either BWA or STAR)

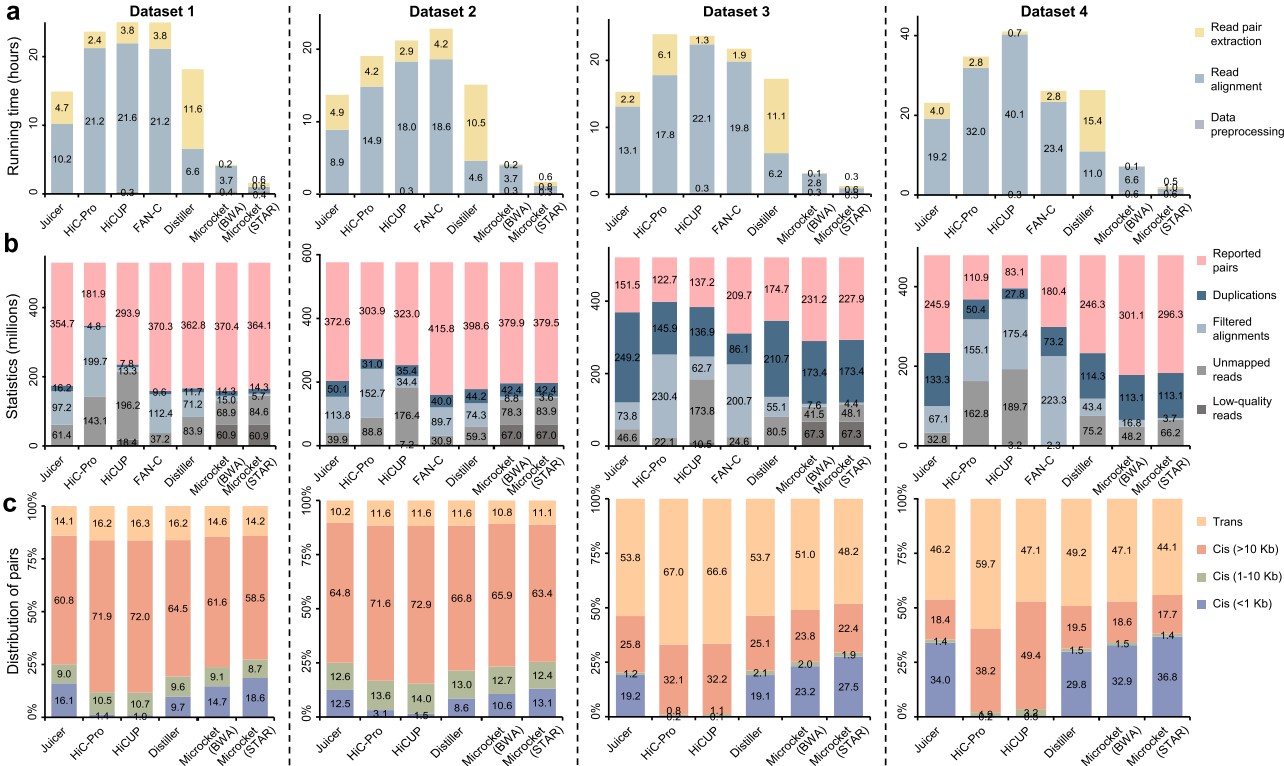

**Fig. 3 | Benchmark evaluations on Hi-C datasets. a** Running time (in hours; averaged from 5 replicated runs); **b** key statistics of the analysis (numbers were in millions); **c** distributions of pairs reported by Juicer and Microcket (using BWA as the aligner). Notably, Microcket supports BWA or STAR as the aligner and results from both aligners were reported; Juicer, HiC-Pro, FAN-C, and Distiller did not incorporate preprocessing procedure (panel **a**), and therefore did not have any low-quality reads in statistics (panel **b**). Filtered alignments were fragments that were reported by the aligner, but were discarded in the downstream stages (e.g., singletons and low-quality mapping results).

versus Juicer or HiC-Pro, paired *t*-tests); the reported pairs of Microcket with BWA showed an accuracy of 99.9%, which was comparable to Juicer and HiC-Pro, and the accuracy for Microcket with STAR was 98.5% (Fig. 2b). Meanwhile, pairs that were only reported by Microcket with BWA or STAR while not in Juicer also showed accuracies of 99.9% and 96.9%, respectively (Supplementary Fig. S2b, c). In addition, we extracted the stitchable fragments (which accounted for ~1/3 of the simulated reads) and re-analyzed the original reads without stitching; as a result, Microcket showed both significantly higher mapping efficiencies and speed in handling the fragments with stitching (Fig. 2c, d; $P = 5.5 \times 10^{-6}$ and $1.2 \times 10^{-9}$, respectively, paired *t*-tests), which justified the effectiveness of our read-stitching strategy.

## Benchmark evaluation on Hi-C datasets

We then performed benchmark evaluations on ten real-world experiments generated from five research groups using various strategies. The results were summarized in Fig. 3 and Supplementary Figs. S4–S14. For Hi-C (datasets 1 and 2) and derivant protocols (dataset 3), we compared Microcket with six widely used and state-of-art tools: Juicer, HiC-Pro[12], HiCUP[11], FAN-C[10], and Distiller (the current mainstream pipeline used in 4DN project). Dataset 1 was collected from Rao et al.[19] (part of 4DN project) containing ~530 million Hi-C reads for both IMR90 and GM12878 cell lines. For IMR90 cell line (Fig. 3a), using 16 threads, Juicer, HiC-Pro, HiCUP, FAN-C, and Distiller, took 14.9, 23.6, 25.8, 25.0, and 18.2 h, respectively, to generate the pairs; in contrast, Microcket finished up in 4.3 and 1.6 h using BWA or STAR as the aligner, respectively, demonstrating a ~3–6× (using BWA) or ~9–16× (using STAR) speed boost over the other tools. Furthermore, Microcket reported 370.4 million pairs using BWA, which was higher than all the others benchmarked on this dataset (Fig. 3b). The results on GM12878 cell line showed a consistent trend to IMR90

(Supplementary Fig. S4). On dataset 2 from Johnstone et al.[20] containing ~600 million Hi-C reads for three colorectal cancer cell lines (i.e., HCT116, LS147T, and SW480), Microcket consistently showed a ~3–6× (using BWA), or ~6–13× (using STAR) speed boost over the other tools (Fig. 3a, b and Supplementary Fig. S5). Dataset 3 contained promoter-capture Hi-C on human lung and liver tissue (both ~500 million reads) from Jung et al.[21], and Dataset 4 contained human neurons and astrocytes (ranged from ~360 million to ~480 million reads) from Song et al.[22]. Microcket showed consistent speed boosts and reported much more pairs (up to 3×) compared to the other tools on these two datasets (Fig. 3a, b and Supplementary Fig. S6, 7).

In addition, in Song et al. dataset where read stitching was enabled, the proportion of reported pairs were much higher for stitchable reads compared to the unstitchable ones (Supplementary Fig. S8). We also down-sampled the neuron data from Song et al. to investigate the running time characteristics of Microcket. As a result, the total running time of Microcket (as well as its three components) was all linearly correlated with read number (Supplementary Fig. S9). In addition, running these tools with 8 threads showed consistent results (Supplementary Fig. S10); apart from running time, Microcket used a moderate level of memory and less storage compared to the other tools (Supplementary Fig. S10).

## Benchmark evaluation on Micro-C datasets

For Micro-C experiments (datasets 5–8), we compared Microcket with Juicer, HiC-Pro, and Distiller, which had been validated in handling such data (Table 1). On dataset 5 from Krietenstein et al.[6] (part of 4DN project) containing ~800 and ~1400 million Micro-C reads for hESC and HFFc6 cell lines, respectively, Microcket with BWA ran ~3× faster than the other tools (Fig. 4a, b and Supplementary Fig. S11). Datasets 6 and 7 were collected from Lee et al.[23] and Turkalo et al.[24] containing ~180–550 million Micro-C reads

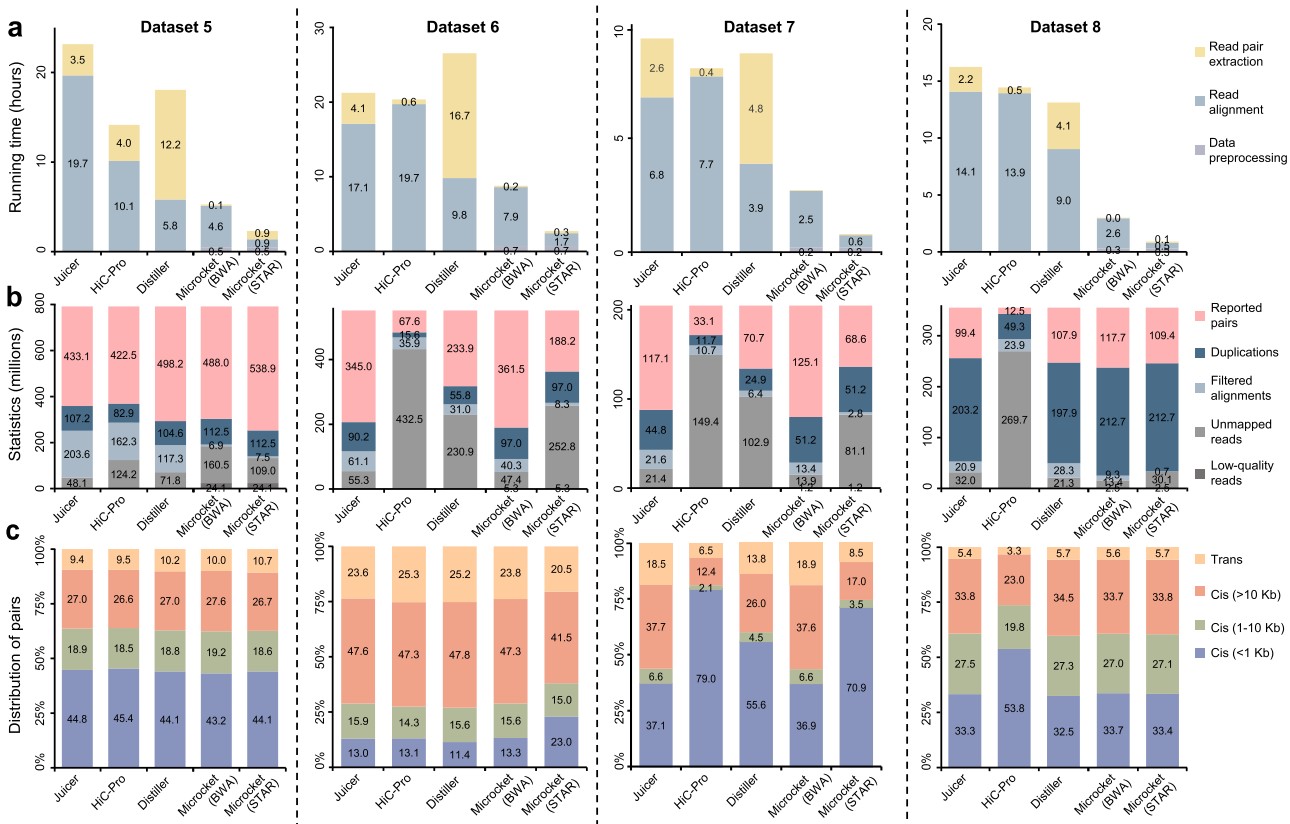

**Fig. 4 | Benchmark evaluation on Micro-C datasets. a** Running time (in hours; averaged from 5 replicated runs); **b** key statistics of the analysis (numbers were in millions); **c** distributions of pairs reported by Juicer and Microcket (using BWA as the aligner).

for various cell lines; Microcket with BWA was also ~3× faster than the other tools, and reported the highest number of pairs (Fig. 4 and Supplementary Figs. S12 and 13). On the last Micro-C dataset with ~350 million reads for K562 cell line generated in this study, Microcket with BWA finished the analysis in ~3 h, which was ~5× faster than Juicer, HiC-Pro, and Distiller, and reported the highest number of pairs (Fig. 4). Notably, 61.3% reads could be stitched in dataset 8, and Microcket reported a higher proportion of pairs on the stitchable reads than the unstitchable ones (Supplementary Fig. S14), which was consistent with the in silico simulation and Hi-C data (Supplementary Figs. S3 and S8).

**Validation on pairs reported by Microcket**

Overall, the reported pairs showed high concordances between Microcket and the other tools (Figs. 3c and 4c). As an illustration, Fig. 5a plotted the Micro-C contact heatmaps derived either from Microcket or Juicer, surrounding the pluripotency gene *SOX2* locus in hESCs. In line with previous findings[25,26], both contact maps revealed *SOX2* resides at the border region of two adjacent insulated neighborhoods and resembled the main features from extra-high depth Micro-C data for hESCs.

In most benchmarked datasets, pairs reported by Microcket were among the most abundant ones, we wondered whether it could help with the downstream functional explorations. To this end, we called chromatin loops in hESC Micro-C data using the results generated by Juicer, HiC-Pro, and Microcket, using the HICUPPS algorithm in Juicer software. Notably, 4DN project provided extra-high depth Micro-C data for this cell line (compiled using 4DN's analysis pipeline, i.e., Distiller) as "gold-standard" to evaluate the performance of these tools. As a result, Microcket recovered more loops, showed much higher recall rate and F-scores than Juicer and HiC-Pro (Fig. 5b and Supplementary Fig. S15). An example was illustrated in Fig. 5c, d: an extra dot (indicated by blue arrow) was identified in Microcket and supported by extra-high depth data, but it was absent in

Juicer. The two anchors of the loop were of high mappability, suggesting that it should not be a false-positive due to mapping errors. In addition, the two anchors were occupied by both CTCF and RAD21 (Fig. 5d), in congruent with the central role of CTCF/cohesin in mediating loop formation.

**Discussion**

The benchmark results demonstrated favorable performance and flexibility of Microcket on 3D genomics data generated from Hi-C and Micro-C protocols, in terms of both speed and mapping efficiency. Benchmark evaluations showed that Microcket (with BWA) achieved preferable performance in speed, accuracy, and efficiency: Microcket (with BWA) was ~3–5× faster than most of the current tools, while the speed-up for Microcket (with STAR) was ~10–20×; in addition, Microcket (with BWA) reported the highest number of pairs in most datasets. Of note, although both Juicer and Microcket employed BWA as the aligner, Microcket was much faster than Juicer, probably due to optimized scheme and parallelization (in fact, many steps in Juicers are run in single-thread mode). In practice, as most 3D genomics projects generate huge-volume data (e.g., billions of reads) to increase resolution, it may take days or even weeks for the current tools to accomplish pairs extraction, while the speed advantage of Microcket could overcome such bottleneck and promise the researchers to explore the biology of their data within just a few h. Furthermore, the increased mapping efficiency may also help in revealing novel insights that are not readily detectable with the current tools. Hence, Microcket possesses high potential in advancing the analysis of the huge-volume 3D genomics data to facilitate in-depth discoveries in 3D genome. On the other hand, Microcket tends to report more short pairs (i.e., <1 kb); such pairs might not be very informative in 3D analysis, while they could be usable in genome assembly or coverage calculations[27].

In addition, the benchmark evaluations also showed that our read stitching strategy could improve the mapping efficiency (Fig. 2c, d,

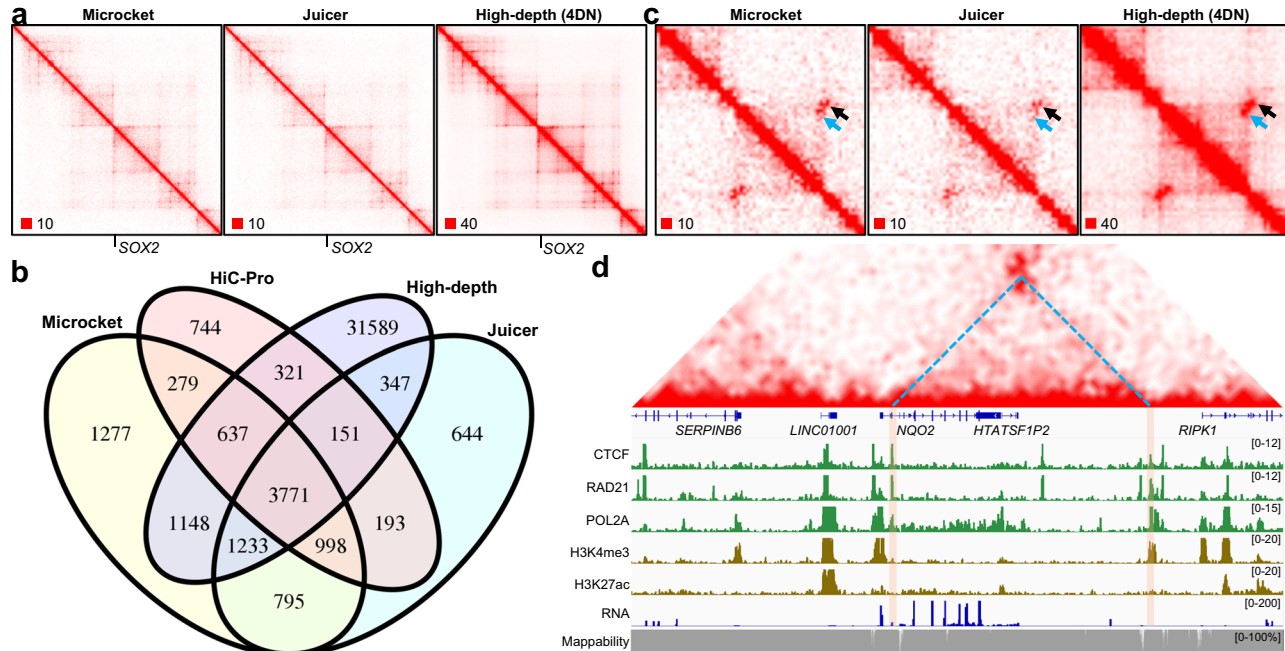

**Fig. 5 | Illustration of loops reported by Microcket. a** Representative Hi-C contact maps using matrices generated by Microcket, Juicer (~800 million reads), and 4DN's high-depth data (~5.9 billion reads) on hESC cell line around *SOX2* gene; **b** Venn gram comparison of the loops reported by Microcket, Juicer, HiC-Pro, and 4DN's high-depth data. **c** A region in chr6p25.2 (coordinate in hg38: chr6:2,950,000–3,080,000). Black arrow highlighted a loop reported in all matrices; blue arrow highlighted a loop reported in Microcket and 4DN's high-depth data, while not in Juicer. **d** Integrative annotations of the region in panel **c** using multi-omics data in hESC cell line, including contact map, ChIP-seq (CTCF, RAD21, POL2A, H3K4me3, H3K27ac), RNA-seq data, and mappability. Dashed lines indicated the loop highlighted by the blue arrow in panel **c**, and brown windows highlighted bindings of CTCF and RAD21 at the loop anchors.

Supplementary Figs. S3, S8, and S14); in fact, we had investigated 10 Hi-C and Micro-C datasets from the recent literature, and we found that stitchable reads accounted for 4.3-85.9% (median: 37.3%; Supplementary Table S1), suggesting that most datasets could benefit from the stitching strategy. Besides read stitching, filtering strategies after alignment also varied among different tools (as shown in "filtered alignments" in Figs. 3 and 4), which might lead to discrepancies in the reported pairs and further investigations are needed to evaluate the performance of the filtering strategies employed among different tools. Considering the high complexity in 3D genome data processing, comprehensive assessments are still needed to fully evaluate the advantages of Microcket in following studies, especially the effects of read stitching strategy, different aligners, as well as accuracy of the extra pairs.

## Methods
### Sequencing reads preprocessing
Raw sequencing reads were first processed by Ktrim[28] to remove adapters and low-quality cycles; reads with identical heading 16 bp in both read 1 and read 2 were considered as PCR duplicates and removed; then the reads were processed by FLASH[29] to identify and stitch those shorter than twice of the sequencing cycles. In current implementation, all three steps were piped to speed up the preprocessing procedure. For read stitch, by default ("auto" mode), Microcket would automatically evaluate the heading 10 K sequencing reads and enable stitch mode if more than 10% reads could be stitched. Of note, the users were allowed to enable/disable stitch mode manually. Based on our evaluations, the proportions of stitchable reads vary from dataset to dataset (Supplementary Table S1), depending on the insert DNA size and sequencing settings.

### Alignment filtering and pairs extraction
After alignment, we first removed all the records with a mapping score lower than 10. Then, the number of mapping records was examined for each read (Supplementary Fig. S2): for stitched reads, we only kept those mapped to

one (i.e., non-chimeric) or two regions (i.e., break into 2 segments); for unstitched reads, besides the two scenarios in stitched reads, the following type of reads was also kept: one compartment was break into two segments, where one segment could be paired to the other compartment (i.e., they were generated from the same fragment). Then, the past reads with more than 50% cycles clipped (i.e., could not be properly mapped) were further removed. Lastly, the two outmost ends of the kept reads were extracted and reported as pairs. The inferred pairs were piped to "sort" program to reduce disk operations and speed up.

### Simulation analysis
The sim3C[30] software was used to generate 10 million paired-end 150 bp in silico Hi-C reads against human genome and MboI enzyme. The mean insert size was set to 300 bp to generate reads that could be stitched (around 1/3 based on our simulation data). The in silico reads were then analyzed by Juicer and Microcket similar to the real datasets. The simulation was repeated for 5 times. To evaluate the accuracy of Juicer/Microcket, extracted pairs with identical chromosomes and the distances between reported and simulated genomic coordinates <500 bp (because sim3C records the enzymatic cutting sites while generate fragment ends nearby to simulate the Hi-C reads) were considered as the correct results. For Juicer, both the raw unfiltered and the filtered pairs (using a mapping quality score of 30) were examined.

### Benchmark evaluations
During the benchmark evaluations, 16 (or 8) threads were used for all software. The reference human genome version used was NCBI GRCh38 (hg38). For Juicer/FAN-C/Distiller, BWA served as the aligner; for HiC-Pro/HiCUP, Bowtie2 served as the aligner. During processing Hi-C data, the digestion-related parameters were manually set to match the experimental protocols of the corresponding datasets for Juicer/HiC-Pro/HiCUP/FAN-C; during processing Micro-C data, the digestion-related parameters were set to "none" for Juicer and were left empty for

HiC-Pro; for Distiller, which did not require digestion-related parameters, all default parameters were used. For Microcket, '-b' option was set for samples in datasets 3 and 8 to activate the "biological replicates" mode of Microcket. Each analysis was repeated for 5 times and the averaged running time was reported. For HiCUP, as the main pipeline stopped after alignment filtering, we adapted its utility program 'hicup2juicer' (provided in HiCUP's source package) to obtain the pairs results. For FAN-C, as it automatically removed the fragment-level pairs in the final results, distributions of the pairs and comparisons to Microcket were both omitted. To assess the concordances between Microcket and Juicer/HiC-Pro/HiCUP/Distiller, mapped reads with identical chromosomes and the distances between reported genomic coordinates <200 bp (as the aligners used clipping and reported different endpoints for each fragment) were considered as the consistent results. Loops in hESC dataset were called using HICUPPS algorithm in Juicer software with a resolution of 5 kb.

### Computational environment

A computing machine equipped with an Intel Xeon Gold 6242 CPU (16-core) and 192 GB memory was used for benchmark evaluations. The operating system of this machine was standard 64-bit Linux (CentOS v7.5.1804, kernel version 3.10.0). For Microcket (v1.3.0 was used in this study), its dependencies (with version information) were as follows: Ktrim[28] v1.5.0, FLASH[29] v1.2.11, and STAR[15] v2.7.9a; for Juicer[9], HiC-Pro[12], HiCUP[11], FAN-C[10], v1.6, v3.0.0, v0.8.3, and v0.9.25, respectively, were used in this study; BWA[14] v0.7.17 and Bowtie2[31] v2.3.5.1 were used as the underlying aligners; for Distiller, pairtools[32] v0.3.0 was used. Other system and dependent software included perl v5.16.3, python v3.10.7, g++ v4.8.5, java v11.0.12, zlib v1.2.7, and samtools[33] v1.12.

### Cell culture and Micro-C experiment of K562 line

K562 cells (ATCC #CCL-243) were maintained in RPMI 1640 medium with 10% FBS and 1× penicillin/streptomycin at 37 °C under 5% $CO_2$. Micro-C libraries were prepared following the procedure as previously described in Krietenstein et al.[6]. In brief, five million K562 cells were first crosslinked with 1% formaldehyde (Sigma, #47608) for 10 min at room temperature and then quenched by 0.75 M Tris pH 7.5. After two rounds of PBS washes, cells were further crosslinked with 3 mM DSG (Santa Cruz, #sc-285455A) for 45 min at room temperature. After quenching with 0.75 M Tris pH 7.5 and the following two rounds of PBS washes, K562 cell pellets were snap frozen in liquid nitrogen and stored at −80 °C. K562 cell pellets were then treated in lysis buffer containing 0.2% NP-40. The extracted nuclei were then digested by 1000 Gel Units of MNase (NEB, #M0247S) at 37 °C for 10 min with 80% to 90% of fragmented chromatin were mononucleosomes. The MNase-digested nucleosomes were then end repaired and incorporated with Biotin-14-dATP (Jena Bioscience, #NU-835-BIO14-L) and Biotin-11-dCTP (Jena Bioscience, #NU-809-BIOX-L). Then the nucleosomes were ligated by T4 DNA Ligase (NEB, #M0202) and the unligated products were removed by exonuclease III (NEB, #M0206). The de-crosslinked DNA was purified by phenol/chloroform/isoamyl alcohol and separated on 6% PAGE gel. A band at the size of 250–400 bp was excised and DNA was extracted and then purified by MyOne™ Streptavidin C1 (Thermo Fisher Scientific, #65001). For the library preparation, end-repair/A-tailing, and adapter ligation were performed on-bead using the VAHTS Universal DNA Library Prep Kit for Illumina V3 (Vazyme, #ND607-01). Ligated DNA was then amplified using 12 cycles on the thermocycler and libraries were purified using the VAHTS DNA Clean Beads (Vazyme, #N411-01). Micro-C libraries were sequenced in paired-end 150 bp mode on an Illumina NovaSeq 6000 sequencer by Genewiz company.

### Statistics and reproducibility

Statistical analyses were performed on R software using paired $t$-tests or linear regressions. For in silico simulation, 5 replicates were performed. For benchmark evaluations, 8 datasets generated using Hi-C or Micro-C protocols were analyzed, and the analyses were repeated 5 times.

### Reporting summary

Further information on research design is available in the Nature Portfolio Reporting Summary linked to this article.

## Data availability

Micro-C data for K562 cell line (dataset 8) has been deposited to Gene Expression Omnibus (GEO) under accession number GSE205500; Hi-C data for GM12878 and IMR90 cell lines (dataset 1) is available from GEO under accession numbers GSM1551583 (the first 3 lanes were analyzed), GSM1551599, and GSM1551600; Hi-C data for HCT116, LS147T, and SW480 cell lines (dataset 2) is available from GEO under accession number GSE133928 (sample ids "HCT116-0h-1_HiC", "HCT116-0h-2_HiC", "LS174T-1_HiC", "LS174T-2_HiC", "SW480-1_HiC", and "SW480-2_HiC" were used in this study); Promoter-capture Hi-C data for human lung and liver tissue (dataset 3) is available from GEO under accession number GSE86189 (sample ids "STL001.LG1" and "STL011.LI11" were used in this study); Promoter-capture Hi-C data for neurons and astrocytes (dataset 4) is available from GEO under accession number GSE113481; Micro-C data for hESC and HFFc6 cell lines (dataset 5) is available from 4D Nucleosome Project under accession numbers: 4DNEXQMEU2O4, 4DNEXC1TYVLD, and 4DNEXXXWOYOB; preprocessed high-depth interaction matrices for hESC cell line is available from 4D Nucleosome Project under accession number 4DNFI2TK7L2F; Micro-C data for C42B cell line (dataset 6) is available from GEO under accession number GSE205000; Micro-C data for human fibroblasts (dataset 7) is available from GEO under accession number GSE212809. The source data behind the graphs in the paper can be found in Supplementary Data 1.

## Code availability

Microcket and scripts used in benchmark evaluations are freely available at https://github.com/hellosunking/Microcket or Zenodo (https://doi.org/10.5281/zenodo.11174864)[34]; Juicer is available at https://github.com/aidenlab/juicer; HiC-Pro is available at https://github.com/nservant/HiC-Pro; HiCUP is available at https://github.com/StevenWingett/HiCUP; FAN-C is available at https://github.com/vaquerizaslab/fanc; Distiller is available at https://github.com/mirnylab/distiller-nf.

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

## Acknowledgements

This work was supported by National Natural Science Foundation of China (NSFC; 32270587, 32100673 to Y.Z., and 82101763 to K.S.), National Key R&D Program of China (2022YFA0912700 to K.S.), Guangdong Basic and Applied Basic Research Foundation (2021A1515012058 to Y.Z., and 2023B1515120073 to K.S.), Shenzhen Science and Technology Program (2023A003, 20231117190144001 to Y.Z.), and Shenzhen Bay Laboratory (to K.S.). We'd like to thank Ms. Qi Wang for technical assistance, and Shenzhen Bay Laboratory Supercomputing Center for computational support.

## Author contributions

Y.Z. and K.S. conceived the study; M.H. and Y.Z. performed experiments; K.S. implemented the software; Y.Z., M.Y., F.G., Y.P., Q.P., L.L., X.L. and K.S. analyzed data; Y.Z., Q.P., L.L., X.L., and K.S. wrote the manuscript; Y.Z., M.Y., F.G. and K.S. revised the manuscript.

## Competing interests

The authors declare no competing interests.
