## [Peer Review File · Communications Biology]

REVIEWERS' COMMENTS:

Reviewer #1 (Remarks to the Author):

I appreciate the effort the authors have made to address my concerns and I am satisfied with the revisions.

It would be interesting to understand the differences in mappability between Microrocket and the other software packages in the simulated data - is there something systematic that Microrocket finds that is not found by the other software tools? Is it the read stitching that does this? And are duplicates only removed at the first pre-alignment stage? Note that Juicer also removes duplicates that align to nearly the exact same position within a pre-defined tolerance due to known issues with Illumina sequencing sometimes skipping nucleotides at the beginning of the read, cf. <https://www.nature.com/articles/s41598-018-31064-7>. If the difference was mostly these types of reads, it's less interesting than if the increased mappability is non-PCR duplicates missed by current technologies.

Point-by-point responses to reviewers' comments

We thank all the reviewers for their valuable comments. Our point-by-point responses to these comments are listed below.

Reviewer #1

I appreciate the effort the authors have made to address my concerns and I am satisfied with the revisions.

It would be interesting to understand the differences in mappability between Microrocket and the other software packages in the simulated data - is there something systematic that Microrocket finds that is not found by the other software tools? Is it the read stitching that does this? And are duplicates only removed at the first pre-alignment stage? Note that Juicer also removes duplicates that align to nearly the exact same position within a pre-defined tolerance due to known issues with Illumina sequencing sometimes skipping nucleotides at the beginning of the read, cf. <https://www.nature.com/articles/s41598-018-31064-7>. If the difference was mostly these types of reads, it's less interesting than if the increased mappability is non-PCR duplicates missed by current technologies.

Response: We thank you for the comments. For simulation data, as the reads were *in silico* generated, there are PCR duplicates, which means the increased mappability of Microrocket indeed benefits from its optimized alignment strategy. The validation using extra-high depth data also demonstrated that the increased mappability of Microrocket is reliable for improved performance in loop calling. In addition, as shown in Suppl. Fig.

S2 (detailed numbers could be found in Supplementary Data), all kinds of pairs (*cis* ($<1K$), *cis* ($1-10K$), *cis* ($>10K$) and *trans*) are increased in Microcket, suggesting that the increased mappability might be general (i.e., does not bias to *cis* or *trans* pairs). Suppl. Fig S3 showed that the stitching strategy could recover $\sim 7\%$ more pairs than non-stitching, which partially contributes to the increase in mappability. Currently, all tools remove PCR duplicates once, and Microcket does this in the pre-alignment stage; however, Microcket reports the results in widely-used *pairs* format, which allows the users to do further filtering (e.g., potential PCR duplicates using Juicer's strategy).